# Industry Scale Optimization: Hammer Crusher and DEM Simulations

Błażej Doroszuk * and Robert Król

Faculty of Geoengineering, Mining and Geology, Wroclaw University of Science and Technology, Na Grobli 15, 50-421 Wrocław, Poland; robert.krol@pwr.edu.pl
* Correspondence: blazej.doroszuk@pwr.edu.pl

**Abstract:** The paper shows the preparation of the numerical models necessary for the simulation mapping of industrial-scale crushers of problematic material, such as copper ore with complex lithology. The crushers investigated in this work are located in the KGHM Polska Miedz S.A. copper ore processing plant. The complex ore consisting of sandstone, dolomite and shale is modeled using the Discrete Element Method (DEM) with Particle Replacement Model (PRM) that was chosen to simulate the crushing process. The article discusses the tests and calibration of material parameters and proceeds to test a breakage model in a laboratory-scale jaw crusher. The results are finally validated with the data from actual industrial-scale crushers and compared with the simulations. As an optimization option, the new shape of hammers is proposed and tested in a numerical environment. The performance of the newly designed hammers was examined using numerical methods. The numerical tests showed that the new design performed worse than the current solution. As a result, time and money were saved by avoiding industrial tests. In conclusion, the work shows how complex processes can be characterized in the numerical environment and used for further analysis.

**Keywords:** hammer crusher; Discrete Element Method; copper ore; breakage model





## 1. Introduction

The optimization of various crusher types is very demanding because many factors influence the final product of the crushing process. The feed parameters (grain size, mass, moisture, rock mechanical properties), the method of filling the crusher, the shape of the working parts, the speed of movement, grate gaps and other parameters all influence the output material. In the case of simple crushers, with jaw or cone, where the gap size and the working frequency can be changed, several mathematical models allow estimations of the efficiency and grain size of the crushed product [1–3].

The most important criterion to be considered when designing a crusher is its failure-free operation. Research on copper ore crushers shows that their components are exposed to damage [4], which can be detected by vibration analysis [5,6]. A breaker failure may cause the processing line to stop working or reduce its efficiency [7].

In the case of crushers which employ the collisions rather than the compression of particles, mathematical modeling of the process has not been a common practice. Individual attempts were made [8] mainly to determine how objects disintegrate during collisions [9,10]. These models were applied mostly to single particles or in laboratory-scale crushing [9]. The performance of impact crushers has usually been empirically determined. The literature indicates that a mathematical model was developed in order to describe the efficiency of shaft impact crushers [11]. However, in the case of hammer crushers, the analysis so far has focused mainly on experiments, most often on a laboratory scale [12] and, in some cases, at an industrial scale, where hammers were analyzed for abrasion [13,14].

Hammer crushers are used in many industries and on different scales [15,16]. In this publication, the focus will be placed on large-scale hammer crushers used in the mineral

processing industry. Such crushers are usually employed during the first stage of the comminution of the mined ore performed in the processing plant. Material several dozen centimeters in size can be broken down to tens of millimeters, e.g., in order to prepare it for further processing in the mills.

The working part of the hammer crusher—the rotor—is most often made of round discs. The discs are mounted on the axis with spaces preserved between them. Hammers are located around the circumference of the discs. They are installed moveably on mandrels so as to rotate freely. When the rotor is set in motion, centrifugal force puts hammers in the working position. On the side opposite to the feeding point, there is a grate whose size matches the desired size of the crushed product. Crushers also often have liners which limit the space between the walls and the rotor so that the crushed material interacts with the hammers more often.

Copper ore mined in Polish mines operated by KGHM Polska Miedź S.A. is characterized by high lithological variability [17]. Depending on the place of extraction, the percentages of rocks in the ore change. The ore is composed of dolomite, shale and sandstone. These rocks have different mechanical and strength properties. Therefore, crushers must be designed to remain operative and operate as efficiently as possible, even when the dominant rock changes in the feed.

The discrete element method (DEM) has been used in the optimization of mining processes for several decades [18,19]. Initially, the optimizations were very simplified, but modern computers have enough computing power to allow attempts at simulating large-scale processes [20]. The method of discrete elements enables simulations of both the behavior of bulk materials and interactions of each particle with other particles and with the elements of the analyzed system separately. The greater number and the smaller size of the particles, the more computing power is needed to perform the simulation. Crushing models that can be used in a DEM environment have also been developed for some time [21]. The models are based on various mechanisms, and in this publication we use the Particle Replacement Model (PRM). The model works in such a way that when the stresses acting on a particle exceed the critical values, the particle is replaced by smaller particles whose number and size depend on the model input data [22].

While other publications focus on individual stages [8–14,20–22] of preparing or using a numerical model, this article describes the entire process from its first stage (sample collection) to its last stage (application of the model). An additional difficulty addressed in this work and not present in other publications lies in the complexity of the lithology of the modeled ore [17], which required the parameterization and the calibration schemes of numerical models to be adapted to this specific case. The article also includes a case study of a specific crusher operating in a processing plant of a copper mine and compares the simulation results with the data obtained from the mine.

## 2. Materials and Methods

### 2.1. Rock Samples

Rock samples were collected at the processing plants of KGHM Lubin, Rudna and Polkowice. The material was sampled by a geologist from the technological line. From each region, approximately 90 $dm^3$ of rock material was collected in the form of fragments approximately 0.5–7 $dm^3$ in size. Macroscopic material samples, identified as sandstone, dolomitic shale and dolomite, were collected on an ongoing basis. In most cases, the qualification of the rock material did not raise doubts, except in the cases when some rock fragments showed substantial surface contamination in the form of dust and sticky damp silty-clay mixture.

As part of the preliminary laboratory work, after washing the surfaces of individual rock fragments, their detailed lithological description was made, followed by the final petrographic identification (Figure 1). The individual samples (rock fragments) were then included in the classes listed in Table 1.

Of all the samples, the material from the Rudna region was characterized by the greatest diversity. During the macroscopic diagnosis, four different types of sandstone were distinguished: weathered, coarse-grained, fine-grained as well as a certain type of sandstone being probably a transitional form with shale. In the samples from the Polkowice region, two varieties of sandstone were observed, differing in firmness. In contrast, there was only one variety in the samples of sandstone collected in the Lubin region.

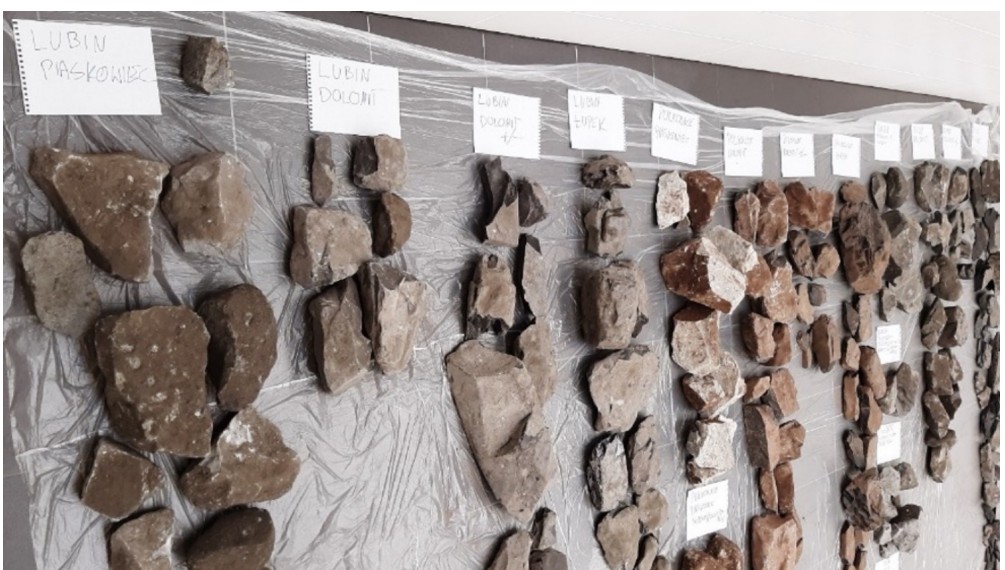

**Figure 1.** Collected samples.

**Table 1.** Rock classes.

| Rock\Region | Rudna | Polkowice | Lubin |
| --- | --- | --- | --- |
| Dolomite | - Calcitic dolomite<br>- Limestone dolomite | - Calcitic dolomite<br>- Limestone dolomite | - Calcitic dolomite<br>- Limestone dolomite |
| Shale | - Clay-dolomitic shale<br>- Dolomitic shale | - Dolomitic shale | - Dolomitic shale |
| Sandstone | - Sandstone 1<br>- Sandstone 2<br>- Sandstone 3<br>- Sandstone 4 | - Sandstone 1<br><br>- Sandstone 2 | - Sandstone |

In order to create a model of the crushing process, it is necessary to know the rocks and their internal structure. Cuts from rock samples from each region were prepared for microscopic analysis (Figure 2). A total of 28 polished thin sections were examined in transmitted light using the Nikon ECLIPSE LV100POL polarizing microscope. The summary of the particle size measurements results is presented in Figure 3, and the averaged values are summarized in Table 2. Initial petrographic identification was also verified.

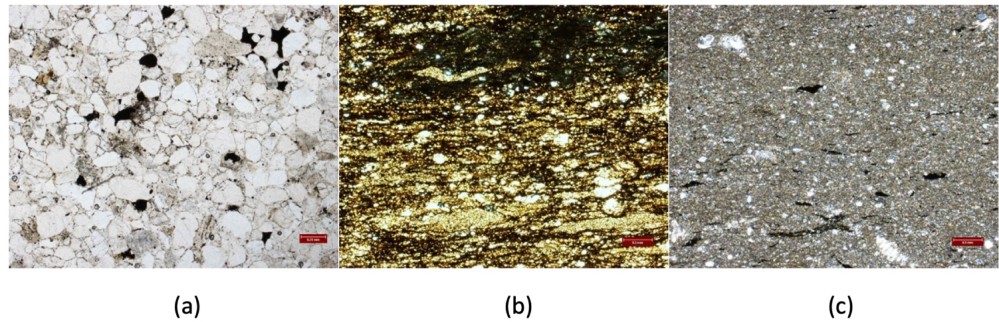

**Figure 2.** Polishes thin sections analyzed under microscope: (**a**) sandstone, (**b**) shale and (**c**) dolomite.

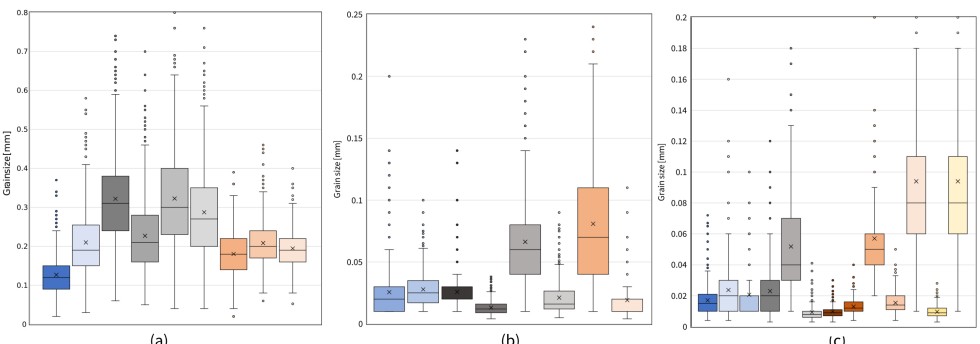

**Figure 3.** Grain size: (**a**) sandstone, (**b**) shale and (**c**) dolomite.

**Table 2.** Average grain size [mm].

| Rock\Region | Lubin | Rudna | Polkowice | Average |
|---|---|---|---|---|
| Sandstone | 0.1689 | 0.2900 | 0.1946 | 0.2178 |
| Shale | 0.0267 | 0.0316 | 0.0500 | 0.0361 |
| Dolomite | 0.0204 | 0.0280 | 0.0275 | 0.0253 |

The grain size of the shale samples was identified by measuring dolomite crystals (microsparite and sparite), which are in the form of lenses and laminates of various sizes or the sizes of aggregates and dolomite nodules, based on microscopic photos.

In the case of three samples of dolomite from the Polkowice region, which had a very complex structure, separate background measurements were made for more minor minerals, and grains were described as "large", i.e., dolomite crystals and aggregates. For samples where background and larger crystals were measured separately, a weighted average was calculated where the background crystals were estimated to be ~70%.

## 2.2. DEM Parametrization and Calibration

The work uses the Discrete Element Method (DEM), which is based on simulations of the movement of separate particles. The calculations performed during the DEM simulation are cyclical. They include calculating the forces acting on the particles and applying Newton's second law of motion to particles to update the acceleration and velocity. After a time step is passed, the position of all particles is calculated and the particle contacts are detected. The last step of the cycle is the calculation of the contact forces [23].

The adopted contact model determines the behavior of the material. In EDEM software, it is possible to use the Hertz–Mindlin (no-slip) contact model. The Hertz–Mindlin contact model is based on the calculation of the normal force $F_n$ according to the Hertz contact theory and the tangential force $F_t$, the calculation method of which is based on the work by Mindlin and Deresiewicz [24]. Both the normal and tangential forces have a damping

component [25], which is related to the restitution coefficient [26]. The tangential friction force is based on Coulomb's laws of friction [25].

The selection of an appropriately small time step is crucial because if a step greater than the critical time step is assumed, it may lead to errors [27]. The value of the critical time step is influenced by the particle size, density and deformation parameters. Many researchers artificially lower the value of the Kirchhoff modulus to shorten the time required to perform the simulation.

When preparing a simulation with discrete elements, two components can be distinguished: particles and geometries. Particles are represented by shape, complexity, moments of inertia as well as mass and volume. The most important parameters characterizing the particles and geometries include Poisson's ratio, density and Kirchhoff's modulus interchangeably with Young's modulus. The most difficult values to choose are the coefficients that characterize the interactions of the particles with the materials of the geometry elements and the interactions of the particles with each other. The creation of a contact model requires three coefficients: restitution, characterizing the loss of velocity after a collision, as well as static and rolling friction. The ore is a mixture of different rocks with different material parameters. Therefore, it is impossible to determine the parameters more precisely than with respect to their ranges or average values.

The static friction coefficient was determined using an inclined plane (Figure 4a). The calculations included the angle of static friction which, if exceeded, causes the material to slide off the plane. Additional samples (Figure 4b) with flat, parallel surfaces were also cut from rocks and cast steel and were placed on the plexiglass to determine the interactions between individual rocks and between rocks and steel.

Each time, ten randomly selected rock particles of a given material were placed in all configurations (material one particle: material two inclined plane liner) and a configuration with acrylic glass for further calibration. The results were then averaged separately for each of the regions, obtaining a set of static friction coefficients: dolomite-shale, dolomite-dolomite, dolomite-sandstone, sandstone-sandstone, sandstone-shale, shale-shale, cast steel-shale, cast steel-dolomite, cast steel-sandstone, plexiglass-dolomite, plexiglass-sandstone and plexiglass-shale, obtaining 12 parameters for each of the three regions.

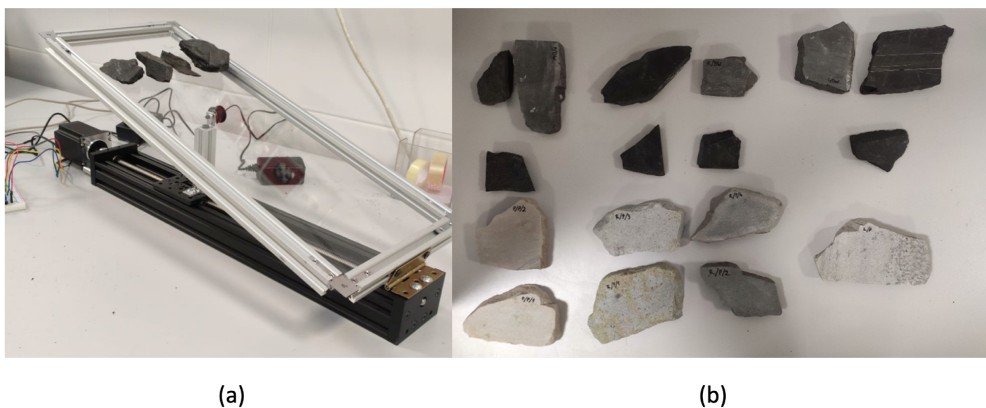

(a)             (b)

**Figure 4.** Measuring static friction: (**a**) incline plane and (**b**) flat surface samples used as liners.

The rolling friction coefficient was not determined because the shape of the solids in the simulation was simplified to single spheres. The coefficient was used to calibrate the behavior of the bulk material.

The coefficient of restitution is a measure of the particle's loss of kinetic energy after the collision [28]. The restitution coefficient is one of the key parameters responsible for the compliance of the DEM simulation with reality. The contact model is used to calculate the velocity of particles after each collision, and simulations can cover thousands of such collisions at the same time. Determining the value of the restitution coefficient is simple in the case of spherical objects. The matter is much more complicated when it comes

to irregularly shaped particles, as in the case of the material excavated from the mine. Irregularly shaped particles behave unpredictably after a collision and have a moment of inertia that is difficult to determine.

Using the principle of the conservation of mechanical energy, many different dependencies can be derived, allowing an indirect calculation of the coefficient of restitution. Using this principle, the position of the particles at their maximum height after reflection was calculated depending on the angle of reflection. For the same restitution coefficient, the points at which the molecules have the highest position during the rebound flight, obtained for different reflection angles in one plane, fit into the ellipse, as shown in Figure 5.

The test was carried out for samples analogous to the static friction coefficient test, obtaining 12 coefficients for three regions. This research was performed with the plotted ellipses for the restitution coefficients from 0.05 to 0.9. Using a high-speed camera, it was observed which ellipse—corresponding to different restitution coefficients—intersects the particle path at the moment when, after reflection, it is at the highest point of its flight.

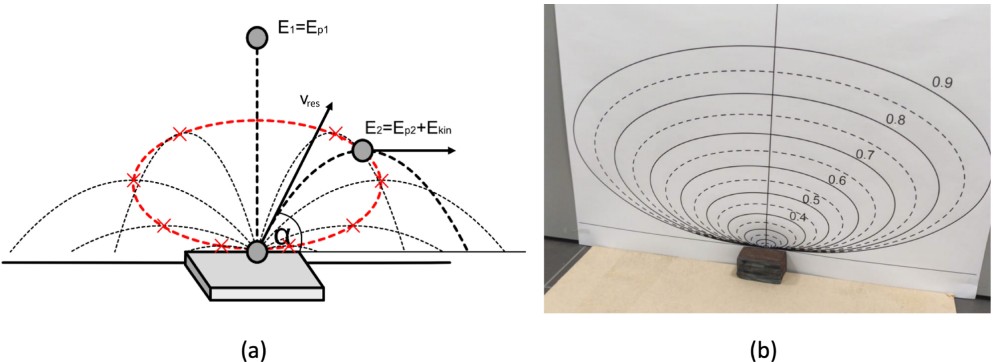

(a)                                   (b)

**Figure 5.** Measuring set for coefficient of restitution: (**a**) idea and (**b**) test stand.

The values of the measured coefficients are summarized in Table 3.

**Table 3.** Measured coefficient of static friction and restitution.

| Pair | Coefficient of Static Friction | | | | Coefficient of Restitution | | | |
|---|---|---|---|---|---|---|---|---|
| | Lubin | Rudna | Polkowice | Average | Lubin | Rudna | Polkowice | Average |
| Dolomite-dolomite | 0.678 | 0.645 | 0.756 | 0.717 | 0.120 | 0.107 | 0.113 | 0.116 |
| Dolomite-shale | 0.600 | 0.653 | 0.734 | 0.662 | 0.075 | 0.130 | 0.139 | 0.115 |
| Dolomite-sandstone | 0.640 | 0.732 | 0.793 | 0.722 | 0.128 | 0.098 | 0.121 | 0.115 |
| Dolomite-plexiglass | 0.509 | 0.480 | 0.450 | 0.480 | 0.065 | 0.123 | 0.080 | 0.089 |
| Dolomite-cast iron | 0.560 | 0.604 | 0.658 | 0.608 | 0.095 | 0.195 | 0.165 | 0.152 |
| Sandstone-shale | 0.688 | 0.785 | 0.787 | 0.754 | 0.075 | 0.132 | 0.105 | 0.104 |
| Sandstone-sandstone | 1.157 | 0.749 | 0.803 | 0.903 | 0.125 | 0.147 | 0.164 | 0.145 |
| Sandstone-plexiglass | 0.503 | 0.531 | 0.632 | 0.556 | 0.115 | 0.074 | 0.108 | 0.099 |
| Sandstone-cast iron | 0.663 | 0.679 | 0.912 | 0.751 | 0.230 | 0.159 | 0.193 | 0.194 |
| Shale-shale | 0.611 | 0.728 | 0.854 | 0.731 | 0.075 | 0.098 | 0.115 | 0.096 |
| Shale-plexiglass | 0.450 | 0.434 | 0.439 | 0.441 | 0.095 | 0.088 | 0.105 | 0.096 |
| Shale-cast iron | 0.472 | 0.600 | 0.615 | 0.562 | 0.105 | 0.125 | 0.185 | 0.138 |

Parameters such as Poisson's ratio, Young's modulus and density cannot be provided directly into the simulation as actual values describing the material. The values adopted for the simulation should be substitute values selected so that

- the actual volume density is consistent with the simulated one (the spherical shape allows another % of the space to be filled because the shape affects the way the particles are packed),

- the forces obtained during the simulation correspond to the real forces transmitted by the particles (for this purpose, the test of uniaxial compression of particles in the cylinder is used) and
- the time step is not too small because Young's modulus influences the necessary time step in the simulation to the greatest extent (therefore, the smallest possible Young's modulus is selected, which ensures a stable simulation and allows a reliable simulation of forces transmitted by particles).

The volumetric density is measured by filling a cylinder with a diameter $d$ = 15 cm and a height $h$ = 15 cm with crushed rocks, successively: sandstone, shale and dolomite. Each of the samples was averaged so that the share of rocks from individual regions was the same. The samples placed in the cylinder of the testing machine were subjected to uniaxial compression tests (Figure 6). The obtained results were used to calibrate the material parameters of the copper ore (Figure 7).

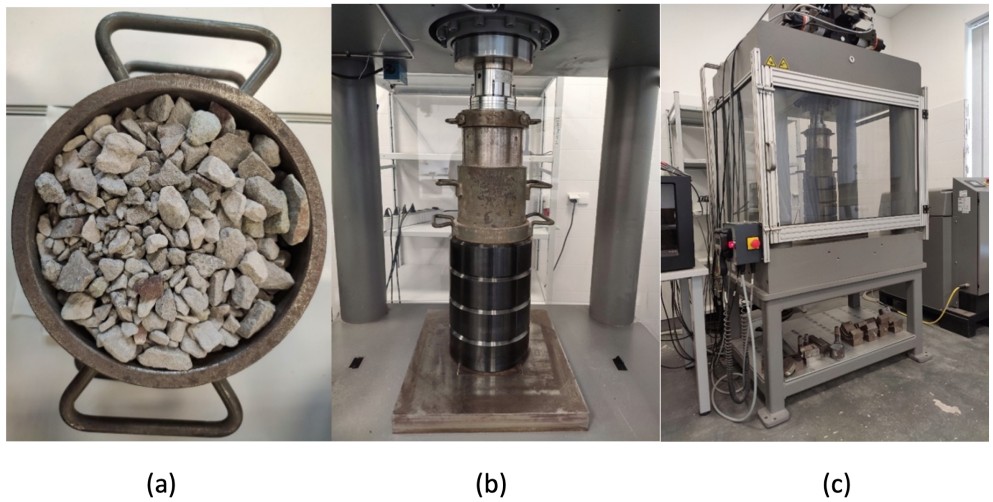

**Figure 6.** Cylinder: (**a**) filled with copper ore, (**b**) placed in the machine and (**c**) during uniaxial compression test.

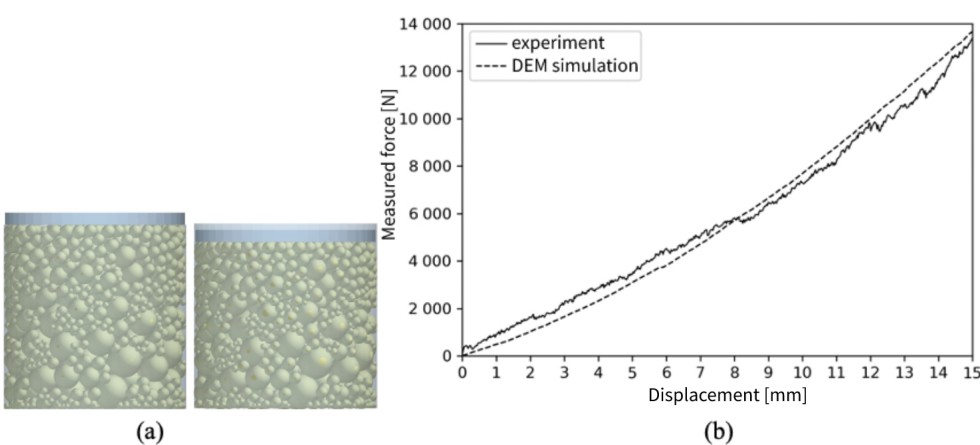

**Figure 7.** Compression of the particles in the cylinder: (**a**) simulation and (**b**) comparison of the registered data.

The test was mapped in the DEM environment, filling the cylinder to the height as during the actual test with the material of the same grain size composition, and then the particle density was selected in the simulation so that the volume density was consistent with the real one. Then, a series of simulations with different values of Young's modulus and Poisson's ratio were carried out, which allowed for the selection of a set of parameters

(Table 4) that best reflect the actual behavior of the bulk material when subjected to a known force.

**Table 4.** Obtained parameters.

| Rock | Density [kg/m$^3$] | Young Modulus [Pa] | Poisson's Ratio [-] | Time Step [s] |
|---|---|---|---|---|
| Sandstone | 2050 | $4.35 \times 10^7$ | 0.25 | $1.00 \times 10^{-5}$ |
| Dolomite | 2130 | $1.38 \times 10^8$ | 0.35 | $1.00 \times 10^{-5}$ |
| Shale | 2100 | $9.30 \times 10^7$ | 0.30 | $1.00 \times 10^{-5}$ |

In the model and simulation tests, a steel sample was used, or rather a Hadfield cast steel with the symbol L120G13 (PN-88/H-83160), characterized by a Poisson's ratio in the range of 0.27–0.30 and Young's modulus equal to approx. 210 GPa. Hadfield cast steel has good mechanical properties. The chemical composition of such steel is about 1C, 12 [%] Mn, and 0.5 [%] Si. Hadfield cast steel is characterized by high ductility; resistance to cracking, impact and abrasion; and is widely used in technological lines, machines and devices used for crushing hard rock. The superior technological properties of the alloy—castability— mean that many structural elements of machines, including the hammers of the crushers adopted for analyses, are manufactured by the casting method and usually used without machining. The undoubted advantage of Hadfield's cast steel is that under pressure and impacts, the crushed layer hardens strongly, while the disadvantage is that under abrasion conditions without high pressures and impacts, the material wears out quickly [29].

The calibration of the model consisted in carrying out a series of simulations that enabled the selection of the contact parameters between the material particles, e.g., sandstone– sandstone, so that the behavior of the simulated material corresponded to the behavior of the actual ore. The work uses the latest methods [30,31] developed by researchers to create a standard test for calibration.

Contact parameters describing interactions between particles of the same material are not real values. They are substitute values compensating for all simplifications of the model in relation to reality. For this reason, it is essential to perform calibration tests comparing the behavior of the simulated material with the actual material.

The calibration stand Figure 8 adopted for the tests consists of two boxes with walls made of acrylic glass, commonly known as plexiglass. There is a locked hole in the bottom of the upper box. After unlocking, the material can freely flow into the lower box.

The work aimed to create a universal model of copper ore mined in KGHM mines, which could approximate ore behavior with variable proportions of rock varieties. Thanks to this approach, the model would be useful in all production regions. Therefore, first, the parameters of each rock variety were averaged with a weight of one-third for the area.

Four experiments were carried out by pouring the crushed material (0.8–1.6 cm fraction) into the calibration box. Separate samples of shale, dolomite, and sandstone were prepared so that one-third of the mass were rocks from each region. One ore sample was also prepared with mixed rocks in the proportions of 40/40/20 (dolomite–sandstone–shale).

The flowing time and the mass poured into the lower box were recorded during the experiments, taking the mass parameter as the main calibration parameter.

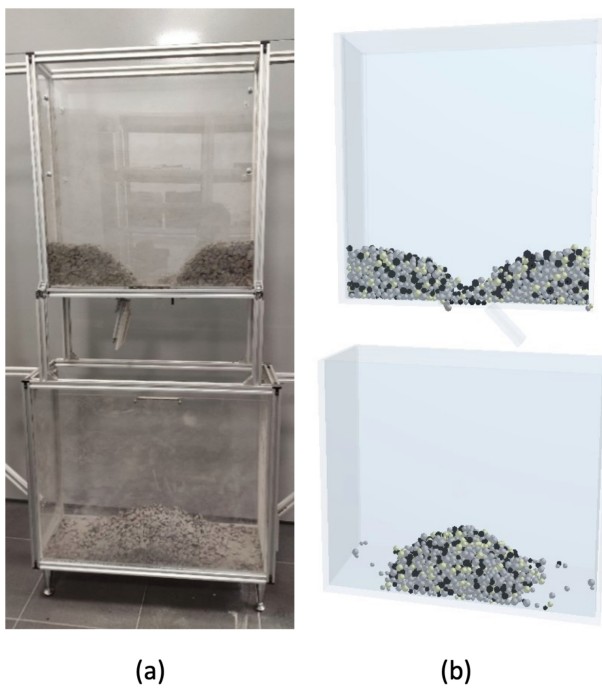

**Figure 8.** Calibration test stand: (**a**) performed experiment and (**b**) simulated experiment.

The parameters presented in the previous tables were adopted for the simulation. The restitution coefficient was assumed to be equal to the one measured during the tests, while calculations were performed for the rolling and static friction coefficients, which have the most significant impact on the material's behavior [30]. The coefficient of the rolling friction between the particles and the geometry elements was assumed to be equal to zero. The coefficient of the rolling friction is used to compensate for the non-spherical shape modeled by spheres, and this parameter is obtained during calibration process as it is impossible to measure. The rolling and static friction coefficient was calibrated separately for each rock variety based on three experiments in the first stage. A series of simulations with combinations of coefficient values in the range 0.1–0.9 were carried out. The final values were those coefficients for which the simulations gave similar values of the pouring time and the poured mass as in the conducted experiments. In the second calibration stage, the calibrated coefficients determining the dolomite–dolomite, shale–shale and sandstone–sandstone interactions were assumed to be constant because the interactions between individual varieties were calibrated. In order to simplify the analysis and reduce the number of variables, the static friction coefficient was adopted as the average of the previously examined coefficients of friction. Only the rolling friction coefficient was calibrated, which was assumed to be the same for all combinations of interactions of different rocks with each other, and during the calibration, the averaged value of the coefficient giving the best results was sought.

Finally, the obtained results ($E$—Young's modulus, $v$—Poisson's ratio, $\varrho$—density, $e$—restitution coefficient, $\mu_r$—rolling friction coefficient, $\mu_s$—static friction coefficient) were collected in Table 5 and accepted for further simulations in the DEM environment.

**Table 5.** Final DEM parameters.

|  | **Sandstone** | **Shale** | **Dolomite** |
|---|---|---|---|
| Materials | $E = 4.35 \times 10^7$ Pa<br>$v = 0.25$<br>$\varrho = 2\,050$ kg/m³ | $E = 9.3 \times 10^7$ Pa<br>$v = 0.3$<br>$\varrho = 2\,100$ kg/m³ | $E = 1.38 \times 10^8$ Pa<br>$v = 0.35$<br>$\varrho = 2\,130$ kg/m³ |
| Sandstone | $e = 0.145$<br>$\mu_s = 0.8$<br>$\mu_r = 0.2$ | | |
| Shale | $e = 0.104$<br>$\mu_s = 0.754$<br>$\mu_r = 0.5$ | $e = 0.096$<br>$\mu_s = 0.7$<br>$\mu_r = 0.7$ | |
| Dolomite | $e = 0.115$<br>$\mu_s = 0.722$<br>$\mu_r = 0.5$ | $e = 0.115$<br>$\mu_s = 0.662$<br>$\mu_r = 0.5$ | $e = 0.116$<br>$\mu_s = 0.6$<br>$\mu_r = 0.4$ |
| Cast iron<br>$E = 2.1 \times 10^{11}$ Pa<br>$v = 0.3$<br>$\varrho = 4\,000$ kg/m³ | $e = 0.194$<br>$\mu_s = 0.751$<br>$\mu_r = 0$ | $e = 0.138$<br>$\mu_s = 0.562$<br>$\mu_r = 0$ | $e = 0.152$<br>$\mu_s = 0.608$<br>$\mu_r = 0$ |

*2.3. Breakage Model Calibration*

While mapping the crushing process in the DEM simulation, it is necessary to implement an additional breakage model. Many models can be used in DEM simulations, but the Particle Replacement Model (PRM) seems to have the most significant potential for industrial applications. The work uses the Tavares Breakage Model implemented in the EDEM environment.

This model registers the different mechanisms of body fracture that occur during particle collisions. It describes the adaptation of the fracture and weakening mechanism through repeated stresses of brittle materials, considering the variability and probability of particle fracture, and reflects the final size distribution of the derived fragments of the material. The nature of the failure mechanisms depends on the material properties and the nature of the external and internal stresses of individual particles. In particular, the model consists of mathematical expressions that describe the critical condition of particle damage, and when this critical condition is met during a DEM simulation, the particle is immediately replaced with a group of smaller-sized particles. In addition, the model's innovative approach that supports particle exchange eliminates potential weight loss.

During crushing, the particles of a given material may suffer internal damage that weakens their structure but does not crack the rock. Stresses lower than required to destroy the particle, if acting repetitively on the particle, lower the energy needed to break it. As a result, after weakening its internal structure, the particle can be broken with the level of stress which was previously insufficient to break it. This effect is called Particle Damage Accumulation [32,33].

After the material is implemented in the DEM environment with a defined set of parameters, the program detects the collisions and calculates the energy absorbed by each particle during the collision. The energy of the collision is compared each time with the assigned destruction energy for a given particle. If the collision energy is lower, the particle will not be destroyed, but the energy needed for it to be destroyed during the subsequent collision will only decrease.

This energy-based model is less sensitive to the choice of the restitution coefficient than other models. The energy loading the particle is a combination of normal energy and tangential energy.

The fragmentation degree of the fragments of the crushed particle is represented by the single parameter $t_{10}$, which describes the proportion of fragments smaller than $1/10$

of the original particle. Derived particles are virtually packed in the mother particle to overlap to some extent, and when the particle is affected by the destructive energy $E_f$, the derived fragments replace the original particle.

The forces acting on the particles in DEM algorithms depend on the degree of overlapping of the spheres, and to avoid the formation of unnaturally large forces when replacing the particles, when these may overlap, global damping strength, global damping time and local damping strength are used. These parameters are responsible for reducing the total force acting on the derived particles for a specified period, and local attenuation determines what part of the energy after the collision is transferred to the derived particles. Correct selection of these parameters is essential for the particles subjected to unrealistic forces resulting from the extensive overlapping of spheres to not reach very high speeds.

Each particle has specific destructive energy assigned on the basis of its size, average destructive energy, and standard deviation. This energy will change depending on the distribution described by the equation [32,34]:

$$P(E) = 1/2[1 + erf(\ln E^* - \ln E_{50}/\sqrt{2}\sigma)] \tag{1}$$

$$E^* = E_{max}E/E_{max} - E \tag{2}$$

where $E$—distribution of fracture energy [J/kg], $E_{max}$—the value of the upper limit of the fracture energy distribution [J/kg], $E_{50}$—distribution median [J/kg] and $\sigma$—distribution standard deviation [-].

The median fracture energy is given by the equation

$$E_{50} = E_\infty/(1 + k_p/k_{st})\,[1 + (d_0/d_p)^\varphi\,] \tag{3}$$

where $E_{\infty,\varphi}$—parameters adjusted to the measurement data [J/kg][-], $k_n$, $k_{st}$—particle and steel hardness [GPa], $d_p$—representative particle size [mm] and $d_0$—the characteristic grain size in the rock [mm].

When the particle is not destroyed, a new destructive energy is calculated from the relationship

$$E'_f = E_f\,(1 - D) \tag{4}$$

$$D = [(2\gamma/(2\gamma - 5D + 5))(eE_k/E_f)]^{2\gamma/5} \tag{5}$$

where $E_f$—fracture energy for a specific particle [J], $eE_k$—effective collision energy [J] and $\gamma$—damage accumulation factor [-].

The Tavares crushing model is mainly based on the most crucial parameter, $t_10$, described by the equation

$$t_{10} = A[1 - exp(-b(eE_k)/E_f)] \tag{6}$$

where $A, b$—parameters adjusted to the measurement data [-], $E_f$—fracture energy for a specific particle [J] and $eE_k$—damage accumulation factor [J].

The larger the $t_{10}$ value is, the finer the derived particles will be. The grain size distribution of the derived grains in the model is based on the standard distributions developed by Tavares in numerous papers [35,36] and is matched to the $t_{10}$ value calculated by the algorithm when replacing the mother particle with derivative particles.

In order to eliminate particle mass losses due to the lack of simulated tiniest fractions after fragmentation, which are often dust, the model includes the so-called "dummy particles"—particles representing the total mass of the tiniest fractions that could not be formed during the breakdown of the parent particle because they were too small. These particles are no longer subject to further degradation and will be treated in the further analysis as part of the tiniest fraction considered.

A uniaxial testing machine was used to obtain more information about the energy needed to destroy the solids (Figure 9). During the test, the force and displacement of the

piston were recorded, enabling the determining of the work performed by the machine destroying the sample, which was equal to the energy of destruction. The total energy supplied to the sample during the test was also determined.

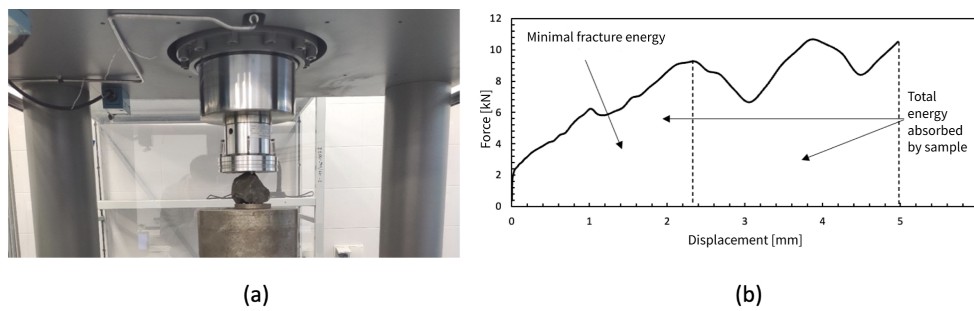

(a)                                                                                     (b)

**Figure 9.** Destroying particles in uniaxial testing: (**a**) sample during test and (**b**) registered data.

Rock samples weighing from 250 g to 5 kg were used for the study. Initially, rock samples randomly selected from among the entire sample set were subjected to uniaxial compression, and then all fragments formed after destruction and having a mass greater than 250 g were compressed once again. The destruction processes of the sample and its fragments were repeated until all of the particles were reduced to less than 250 g. After the destruction process, the samples were screened and the particle size composition was determined. The fragments that weighed more than 10% of the parent particle were few and easy to identify. Their mass was measured each time. Composition of the derived fragments is needed because during the crushing process in the crusher, the rocks are broken not once, but many times, until the desired size is reached.

During the research, together with derivative fragments, 71 shale samples, 83 samples of dolomite and 69 sandstone samples were destroyed. The parameters were averaged, first for samples of the same macroscopically recognized variety (e.g., clay dolomite and limestone dolomite), then all sandstone, dolomite and shale varieties were averaged within each region. The values of the parameters from individual regions were used, after averaging, to develop a general crushing model for the copper ore from KGHM. The data obtained during the study and the literature data on petrographically similar rocks, which were already modeled using the Tavares Breakage Model [35,36], allowed the adoption of a set of parameters describing the crushing of individual lithological varieties of copper ore, which are included in the following Table 6.

**Table 6.** Breakage model parameters.

| Symbol | Parameter | Sandstone | Shale | Dolomite |
|---|---|---|---|---|
| $\gamma$ | Damage constant | 5 | 5 | 5 |
| Global Damping Time | Duration of global damping strength upon the particle [s] | 0.5 | 0.5 | 0.5 |
| Global Damping Strength | Percentage of total contact force upon the particle [%] | 0.5 | 0.5 | 0.5 |
| Local Damping Strength | Fraction of overlap between fragments | 1 | 1 | 1 |
| $E_\infty$ | $E_{50}$ parameter [J/kg] | 23.61 | 92.66 | 76.19 |
| $d_0$ | $E_{50}$ parameter [mm] | 0.218 | 0.036 | 0.025 |
| $\varphi$ | Fitting parameter | 1.2 | 1.2 | 1.2 |
| $\sigma$ | Standard deviation of the fracture energy | 0.4 | 0.3 | 0.3 |
| $A$ | Impact breakage parameter used in the calculation of the $t_{10}$ | 33.9 | 29.3 | 27.9 |
| $b$ | Impact breakage parameter used in the calculation of the $t_{10}$ | 0.03 | 0.02 | 0.02 |
| $d_{min}$ | Minimum particle size for breakage [mm] | 3 | 3 | 3 |
| $E_{min}$ | Minimum collision energy [J] | $1.0 \times 10^{-3}$ | $1.0 \times 10^{-3}$ | $1.0 \times 10^{-3}$ |
| $C_t$ | Fraction of shear energy | 0.2 | 0.05 | 0.1 |
| Truncation ratio | Upper limit of fracture energies | 100 | 100 | 100 |

## 3. Results of Numerical Analyses

### 3.1. Laboratory Scale

The LAB-03-260/S jaw crusher EKO-LAB was used to validate the crushing model. The jaw was set to move in the range of approx. 32.5 to 46.5 mm. The elements of the crusher interior have been mapped in a simplified CAD model (Figure 10).

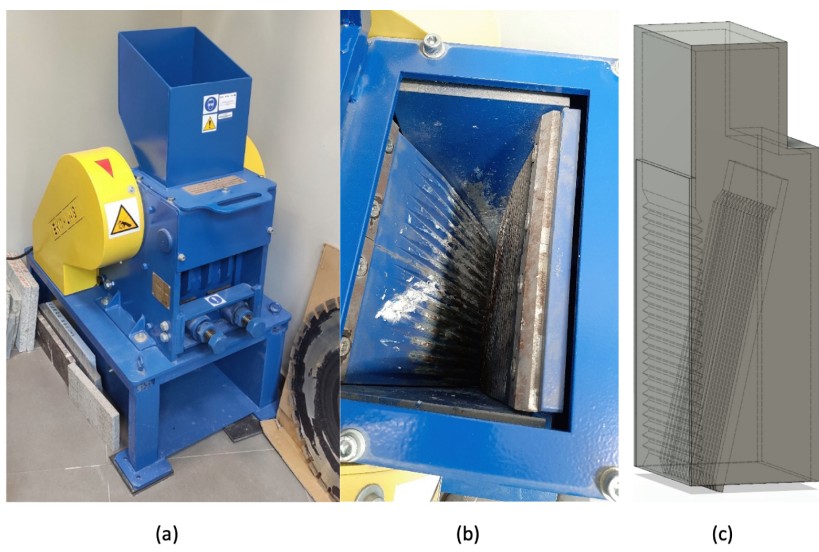

(a)                                           (b)                                           (c)

**Figure 10.** Laboratory scale crusher: (**a**) jaw crusher, (**b**) jaws and (**c**) CAD model.

Four crushing processes were carried out for averaged shale samples, dolomite and sandstone, and mixed ore. In total, 15.1 kg of shale, 12.2 kg of sandstone, 9.2 kg of dolomite and 25 kg of mixed ore were used for the tests. The process of crushing the same amount of material was mapped in the DEM environment, successively for each lithological variety and the mixed ore.

In the simulations, the grain composition of the feed for crushing was assumed for the exact percentages of individual classes as the actual feed. Each class in the simulation was represented by the average grain in class—that is, by spheres with a diameter equal to the average value from the lower and upper limits of the grain class. The grain composition curve of the actual feed and crushing products in the laboratory crusher and the simulation test conducted are presented in Figure 11.

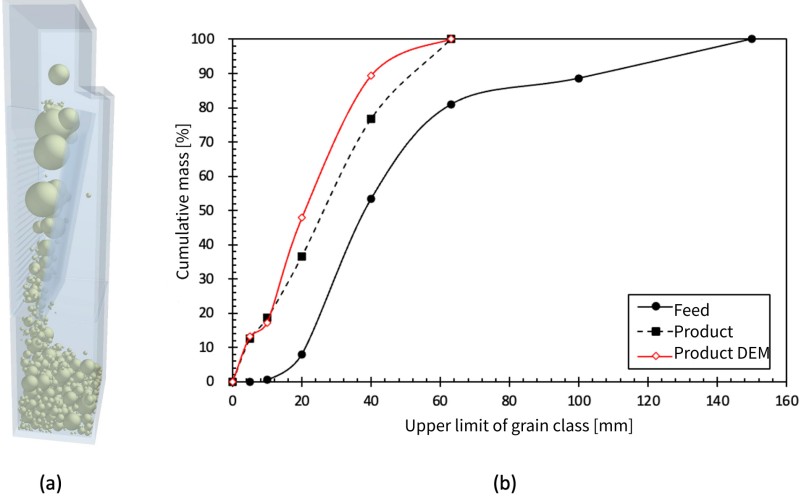

(a)                                           (b)

**Figure 11.** Simulation of copper ore crushing: (**a**) DEM environment and (**b**) obtained data compared with real experiment.

The similarity of the grain curves for both the simulated and the actual crushed products, obtained for identical technical and technological parameters (adopted for crushing in laboratory conditions), confirmed that multiparametric DEMs and PRMs of copper ore can be potentially used to map the process with sufficient accuracy.

### 3.2. Industrial Crushers

The provided documentation allowed the construction of a CAD model of the hammer crusher, i.e., MAKRUM 40.80 (Figure 12). The model has been simplified to be compatible with the DEM simulation environment. The shape of the hammers was unchanged, keeping the geometry in line with the documentation provided.

The simulations were perform on the basis of the obtained numerical models, CAD model, and the available data on the quantity and grain composition of the feed going to the crusher. They represented the operation of the crusher under typical conditions (Figure 13).

The grain composition of the feed was adopted on the basis of the data from the processing plant. All classes below 10 mm were accumulated into one class of 0–10 mm, with a total share of 8.2% and an average grain of 5 mm. 300 mm was assumed as the upper grain size of the largest fraction, in line with the maximum size of pieces that can enter the processing plant.

The obtained results (Figure 14) were compared using the average (50%) size of the particles of the final crushing product and the value of the diameter where 80% particles are smaller. Additionally, the evaluation was based on the primary index describing the crushing process, called the fragmentation degree, defined as the ratio of the representative particle size of the feed $D$ to the particle size of the product $d$ (Table 7).

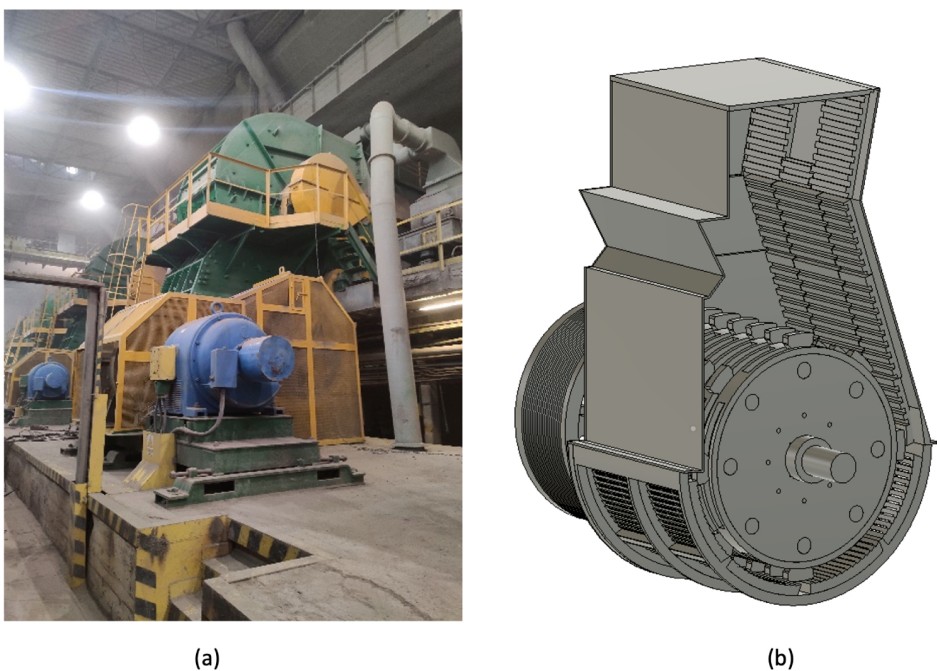

(a)                                                                 (b)

**Figure 12.** Hummer crusher MAKRUM 40.80: (**a**) in processing plant and (**b**) CAD model.

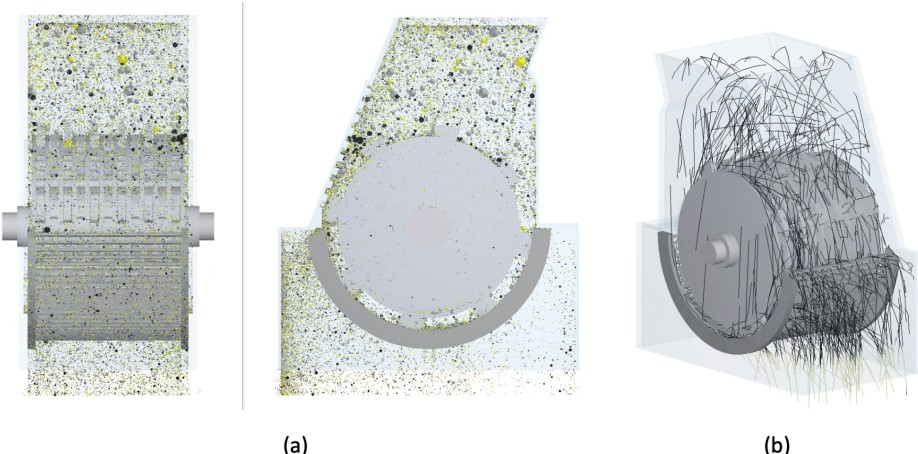

Figure 13. Simulation of the crushing: (**a**) particles inside the crusher and (**b**) particle trajectories.

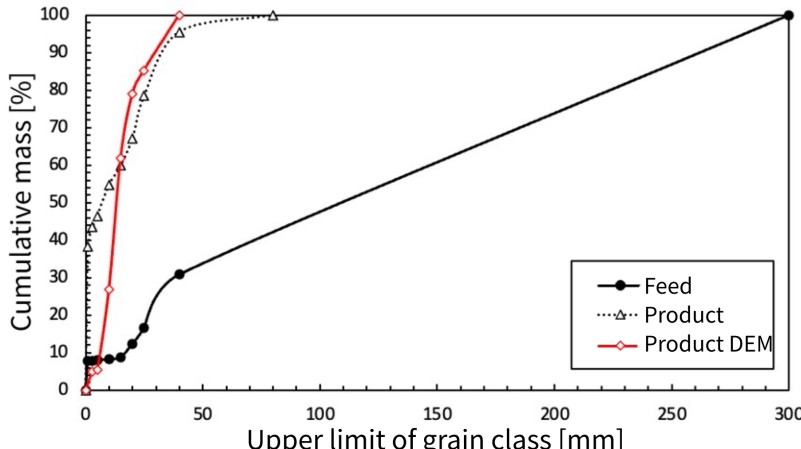

**Figure 14.** Particle size distribution of the feed and crushing product.

**Table 7.** Fragmentation degree.

| Parameter | Feed | Product | Product DEM |
|---|---|---|---|
| $d_{50}$, mm | 109.3 | 7.1 | 13.2 |
| $d_{80}$, mm | 223.8 | 26 | 20.5 |
| Fragmentation degree $D_{50}/d_{50}$ | | 15.39 | 8.28 |
| Fragmentation degree $D_{80}/d_{80}$ | | 8.61 | 10.92 |

Compared to the actual product, each grain composition curve of the DEM product was produced by averaging the results of the tests carried out for three simulations.

### 3.3. Testing Other Hammer Shapes with Prepared Model

Designing, manufacturing and testing new hammers in actual conditions is a lengthy and costly process. Therefore, the use of the prepared model to try new hammer shapes is the optimal solution. The aim of this study was to check how the use of lighter single-sided hammers would affect the crushing process.

New single-sided hammers were designed on the basis of a simplified stress analysis in the FEM environment (Figure 15a). The inside of the mounting ring was assumed in the analysis as a fixed element, and the load was applied perpendicularly to the hammer face. Then, the parts of the hammer that transferred the least stress were removed from the upper part so that the remaining mass was 60% of the initial mass (Figure 15b).

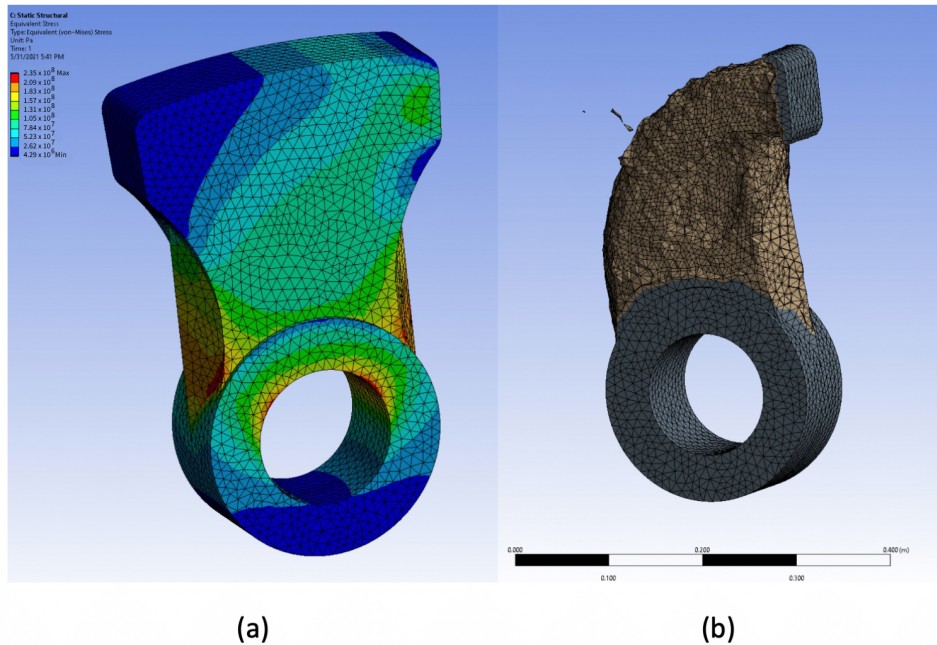

(a) (b)

**Figure 15.** Hammer: (**a**) MES analysis and (**b**) mass reduction.

The resulting body was encased to concentrate the mass around the stress-transmitting areas (Figure 16a). The resulting hammers were replaced in the CAD model of the rotor used in the simulations (Figure 16b).

It was observed that the change of the hammers did not affect the grain size of the crushed product grains. Therefore, to compare the new solution, an analysis of the mass of ore particles inside the crusher was performed. The mass was calculated, as the masses of both particles appearing in the system and the particles disappearing from the system were known (Figure 17). Based on the simulations, it was observed that the mass inside the system stabilizes after some time and no longer shows an increasing trend. The measurement was adopted as the stabilization moment, after which five successive measurements at 0.5 s intervals are indicated by a lower value of the total mass inside the system.

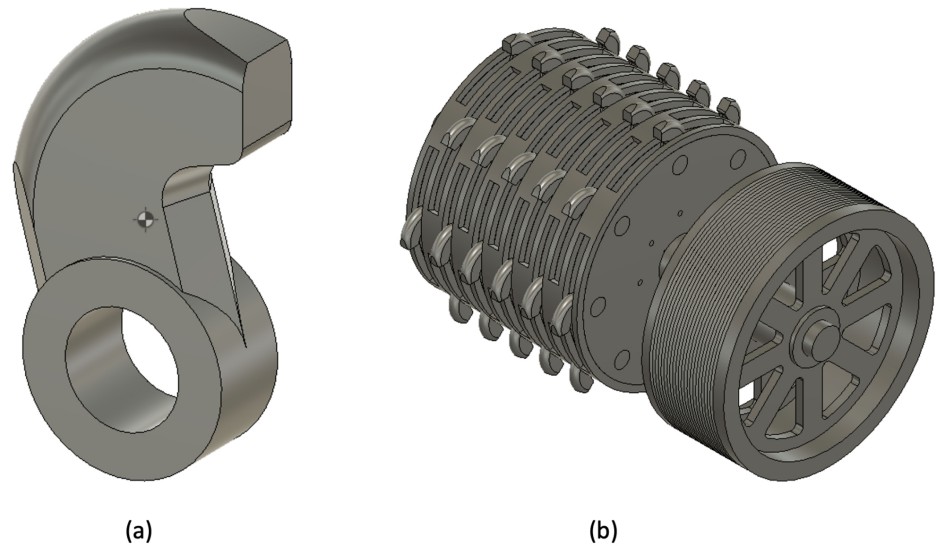

(a) (b)

**Figure 16.** Single-side hammer: (**a**) CAD model of hammer and (**b**) CAD model of rotor with single-side hammers.

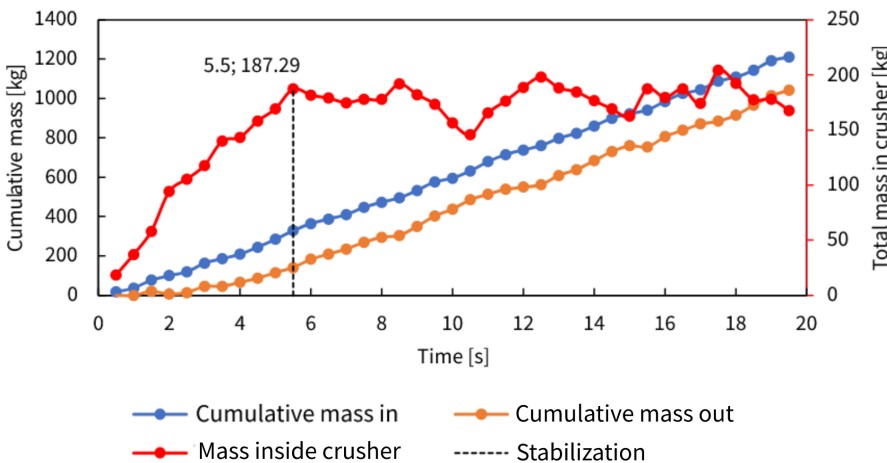

**Figure 17.** Mass inside the crusher during simulation.

The mass in the crusher when its operation stabilized under normal conditions was taken as the reference mass. For the simulation with single-sided hammers, a graph was made (Figure 18) showing the mass inside the system as a percentage of the reference mass. It has been observed that the use of single-sided hammers significantly prolongs the stabilization of the mass inside the crusher, which is probably due to the extension of the time needed to crush the material to a fraction that can pass through the grate. This is also confirmed because the total mass after stabilization is almost 2% higher than the reference mass. More material accumulates inside the crusher due to slower crushing.

Performing analyses in a simulation environment saves a lot of the time and money needed for large-scale industrial tests using the trial-and-error method to assess the suitability of the solution.

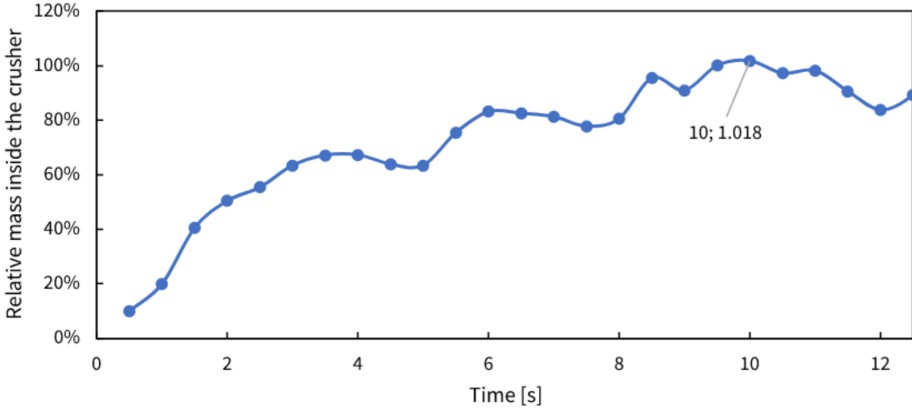

**Figure 18.** Relative mass inside the crusher during simulation with one side hummers.

## 4. Discussion and Conclusions

In order to analyze the phenomena of ore crushing in hammer crushers, digital models of the following studied objects were built: rocks with different lithology and granulation, crushing machines and the crushing process itself.

In order to test a wide range of copper ore rocks found in KGHM mines, rock samples were collected and cataloged from the feed to the processing plants. Their parameters were identified by laboratory methods appropriate for petrographic analyses. Critical parameters for modeling the crushing process: static friction angle, restitution coefficient, density, Young's modulus, Poisson's ratio and rock crushing energy were determined in laboratories at special stands, calibrated or selected with reference to data from the literature. Significant differences in rock parameters from different regions were identified.

In order to validate the developed digital models of operations related to pouring bulk materials (in the DEM environment), auxiliary physical and digital models of basic material flow were built. This allowed the calibration of the DEM models of the studied ore.

Models of engineering objects—the crushers themselves and their elements in direct contact with the solids of crushed ore—were constructed in the digital DEM environment using complex CAD spatial models and considering their material parameters. It was necessary to recreate the analog documentation. Parameterized digital models allow a relatively quick modification of key design features.

As a result, a comprehensive digital copper ore crushing model was built. It allowed a map of the phenomenon of particle collision and their fragmentation occurring during crushing—the formation of derivative fragments and compression, which in total lead to the replacement of the set of primary particles (feed) with the set of particles subjected to crushing (the product). The feed and the product size distribution curves are measurable characteristics, allowing a comparative analysis of numerical experiments and industrial measurements.

The usefulness of the model in carrying out analyses of modifications to the crusher elements was demonstrated on the example of the implementation of alternative hammers on the rotor.

**Author Contributions:** Conceptualization, B.D. and R.K.; methodology, B.D.; software, B.D.; validation, B.D.; formal analysis, B.D.; investigation, B.D.; resources, B.D. and R.K.; data curation, B.D.; writing original draft preparation, B.D.; writing review and editing, R.K.; visualization, B.D.; supervision, R.K.; project administration, R.K.; funding acquisition, B.D. and R.K. All authors have read and agreed to the published version of the manuscript.

**Funding:** The research work was partly co-founded by KGHM Polska Miedź S.A. (agreement no. KGHM-ZW-U-0051-2020) and the research subsidy of the Polish Ministry of Science and Higher Education granted for 2021.

**Institutional Review Board Statement:** Not applicable.

**Informed Consent Statement:** Not applicable.

**Data Availability Statement:** The data presented in this study are available on request from the corresponding author.

**Acknowledgments:** We want to acknowledge cooperation with the processing plants of KGHM mines. The management provided us with samples and documentation necessary for the analysis.

**Conflicts of Interest:** The authors declare no conflict of interest. The funders had no role in the design of the study; in the collection, analyses or interpretation of data; in the writing of the manuscript; or in the decision to publish the results.

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
