# Peer review of "Industry Scale Optimization: Hammer Crusher and DEM Simulations"

_minerals, doi:10.3390/min12020244_

Round 1

Reviewer 1 Report

Reviewer’s comments: 

In this paper, authors model the operation of a hammer crusher using DEM. The authors achieved their objective function through characterizing the rock samples used, determining the DEM parameters and eventually using the Particle Replacement Model to simulate the crusher. The paper has technical merit but it is poorly organized and the introduction is poorly written. The authors should organize the paper into proper sections which includes the methodology, results and discussion, and conclusion as per the journal’s guidelines. Thus, in its current state,  the paper is unpublishable.

The issues found are listed below.

Abstract

  • The first statement needs revision.
  • Line 5 – it appears as if the authors are claiming that the Particle Replacement Model is the only model that can simulate crushing, therefore, the statement should be revised.
  • It is the experimental work that validates simulation work not the other way round.

Introduction

  • The introduction is not actually describing the problem statement as is expected. In the abstract, the authors started by highlighting the key objective of the study, which is to address problems encountered in the crusher optimization schemes that use numerical methods. This thus raised my expectations to see a good review of the different optimization methods used and the problems encountered using numerical methods to optimize crushers. Then the authors’ point of departure from what is currently known in literature, proposing their approach and how it would address the problems mentioned in the review. That way, the contribution of the work to the body of knowledge would have been clearly shown. In its current state, the introduction is poorly written and lacks proper arguments to justify the importance of the work to the body of knowledge.
  • Line 14 – ‘on a different scale’ should be replaced by ‘at different scales’
  • Line 16 – this statement is wrong since the first stage of comminution is blasting.
  • Lines 19-21 – the sentence needs revision.
  • Lines 29-31 – is it to remain operative or to operate efficiently?

Model Preparation

  • Line 95 – It should be “The selection of an appropriately small…”
  • Line 110 – the word “otherwise” is wrongly used.
  • Lines 176-182 – The paragraph is a repetition of the one preceding it.
  • Line 233 – In order to simplify the analysis not the analyzes
  • Table 5 – definition of the symbols is required.
  • Line 239 – remove ‘the’ before the word Table 5.
  • Lines 260-262 – the statement needs revision.
  • Line 307 – change analyzes to analysis.
  • What do authors mean by saying the mass greater than 10% of the mother particle mass was recorded? How is that possible?

Simulation of Crushing

  • Lines 349-352 – To claim that there is high convergence in Figure 11 between the experimental and the DEM products, is misleading. At some points the difference is around 10 %. The authors should statistically validate their claim or rather tone down their claim.

Testing other hammer shapes with prepared model

  • Line 387 – did the authors mean the grains of the crushed product?

Results and Summary

  • Lines 633-635 – the level of accuracy claimed by the authors is not observed in the presented graphs. That kind of claim needs to be backed by statistical validation.

Reviewer 2 Report

The paper addresses problematic of industrial-scale hammer crushers optimization with numerical methods. The whole work is complete and interesting. However, I have some concerns about the details.

Please note my comments below:

  1. In the abstract, I cann’t understand “…showing that new hammers perform worse than actual ones.”
  2. In Fig. 1, we can see that all ore particles are non-spherical. However, the particles used in the DEM simulation are spherical. So, why not use non-spherical discrete element for simulation? That should be more accurate.
  3. “In order to create a model of the crushing process, it is necessary to know the rocks and their internal structure.” How does the internal structure of these rocks guide the modeling?
  4. Figure 3 is difficult to understand. What does the color in the figure represent?
  5. Can the device in Figure 4 measure the friction coefficient between ores? I think it can only measure the friction coefficient between ore and glass.
  6. Can the device in Figure 5 measure the restitution coefficient between ores? I think the restitution coefficients listed in Table 3 are so small.
  7. Only qualitative results are shown in Figure 8. Is there any quantitative data?
  8. What does the rolling friction coefficient in Table 5 mean? How to determine the rolling friction coefficient when spherical particles are used to represent non spherical particles?
  9. Why not compare the power consumption in both simulation and experiment of the crusher?
  10. Can figure 13 (b) give any meaningful information?
  11. The angles of views (a) and (b) in Figure 15 should be consistent.

Round 2

Reviewer 1 Report

The authors have improved the introduction but it still lacks strong arguments which highlights the contribution of the work to the body of knowledge. The authors have also been repeating or leaving some corrections suggested not corrected through the paper. A case in point is the use of the word analyzes. Although the authors have revised other statements at my request, some of the revised statements still needs further improvement to easily understand them. Examples are points 10 and 16 of the first reviewer among other statements in the cover letter. Moreover, the added paragraphs in the Introduction Section are littered with grammatical errors. I thus propose the authors to enlist the service of an English editor before the paper can be accepted for publication.

Abstract

The authors modelled the crushing of copper using DEM and PRM. They tested the model at laboratory scale using the jaw crusher and at industrial crusher using the hammer crusher. The later test gave the authors confidence to propose the new shape of hammers. If this is the case, I do not see how the paper is addressing problems encountered in the preparation of crusher optimization models. The authors followed a modelling scheme routine used in previous research about the subject. Thus, they should revise the first statement in the abstract.

Introduction

In their review of the published literature about hammer crushers, the authors claim that the hammer crusher analysis done so far was experimentally based with most of them done at laboratory scale and few at industrial scale but focusing on abrasion. In this present work, the authors mentioned that they are focusing on hammer crushers at industrial scale but failed to specify their focus area which contributes to the body of knowledge, thus highlighting the importance of the work. The introduction did not review published work in the direction of model optimisation problem solving as promised in the abstract. Neither was it observed in the subsequent sections. It therefore has to be improved.

Results of Numerical Analysis

In the last paragraph of Section 3.1, the reviewer disagrees with the use of the word convergence to describe the plots in Figure 11 and also the claim that multiparametric DEM and PRM models of copper ore can map the process with enough accuracy, based on the presented results in this work. The authors really need to tone down their claim.
